# Role of Endoscopic Ultrasonography-Guided Fine Needle Aspiration/Biopsy in the Diagnosis of Autoimmune Pancreatitis

**DOI:** 10.3390/diagnostics10110954

**Published:** 2020-11-15

**Authors:** Kensaku Noguchi, Yousuke Nakai, Suguru Mizuno, Kenji Hirano, Sachiko Kanai, Yukari Suzuki, Akiyuki Inokuma, Tatsuya Sato, Ryunosuke Hakuta, Kazunaga Ishigaki, Kei Saito, Tomotaka Saito, Tsuyoshi Hamada, Naminatsu Takahara, Hirofumi Kogure, Hiroyuki Isayama, Kazuhiko Koike

**Affiliations:** 1Department of Gastroenterology, Graduate School of Medicine, The University of Tokyo, Tokyo 113-8655, Japan; noguchik-int@h.u-tokyo.ac.jp (K.N.); mizunos-int@h.u-tokyo.ac.jp (S.M.); kanais-int@h.u-tokyo.ac.jp (S.K.); suzukiyu-int@h.u-tokyo.ac.jp (Y.S.); inokumaa-int@h.u-tokyo.ac.jp (A.I.); satotat-int@h.u-tokyo.ac.jp (T.S.); hakutar-int@h.u-tokyo.ac.jp (R.H.); ishigakika-int@h.u-tokyo.ac.jp (K.I.); saitoke-int@h.u-tokyo.ac.jp (K.S.); saitot-int@h.u-tokyo.ac.jp (T.S.); hamadat-int@h.u-tokyo.ac.jp (T.H.); takaharan-int@h.u-tokyo.ac.jp (N.T.); kogureh-int@h.u-tokyo.ac.jp (H.K.); koike-1im@h.u-tokyo.ac.jp (K.K.); 2Department of Endoscopy and Endoscopic Surgery, The University of Tokyo Hospital, Tokyo 113-8655, Japan; 3Department of Gastroenterology, Tokyo Takanawa Hospital, Tokyo 108-8606, Japan; khirano-tky@umin.ac.jp; 4Department of Gastroenterology, Graduate School of Medicine, Juntendo University, Tokyo 113-8531, Japan; isayama-tky@umin.ac.jp

**Keywords:** autoimmune pancreatitis, endoscopic ultrasonography, fine needle aspiration/biopsy, diagnosis

## Abstract

Type 1 autoimmune pancreatitis (AIP) is histologically characterized by lymphoplasmacytic sclerosing pancreatitis (LPSP). Recently, the diagnostic yield of endoscopic ultrasonography-guided fine needle aspiration/biopsy (EUS-FNA/B) for AIP has been reported. However, its role in the diagnostic flow of AIP is not fully elucidated. We retrospectively reviewed 53 consecutive patients who were suspected with AIP and underwent EUS-FNA/B. We evaluated the contribution of EUS-FNA/B to the diagnosis of AIP before considering response to steroid therapy among patients with diffuse enlargement of the pancreas and those with focal enlargement, respectively. Twenty-two patients showed diffuse pancreatic enlargement and 31 showed focal enlargement. The final diagnosis was definitive AIP in 32 patients, probable AIP in 2, possible AIP in 1, and mass-forming focal pancreatitis in 18. There was no change in diagnosis after EUS-FNA/B among patients with diffuse pancreatic enlargement, while diagnosis changed in 38.7% (12/31) among those with focal enlargement—there was a probable to definitive diagnosis in 4 patients, unspecified to definitive in 3, and unspecified to probable in 5. EUS-FNB provided a significantly higher sensitivity for typical pathological findings of LPSP than EUS-FNA, and 10 patients were diagnosed as pathologically definitive AIP by EUS-FNB, though none were by EUS-FNA (*p* = 0.002). EUS-FNA/B was useful in the diagnosis of focal type AIP, and steroid therapy could be introduced after the diagnosis was confirmed. Meanwhile, EUS-FNA/B provided no contribution to diagnosis of diffuse type AIP. EUS-FNB showed a higher diagnostic yield than FNA.

## 1. Introduction

Autoimmune pancreatitis (AIP) is a particular form of pancreatitis with suspected involvement of autoimmune mechanisms in its pathogenesis [1,2,3,4]. AIP is classified histologically as type 1 and type 2, and type 1 AIP is characterized by lymphoplasmacytic sclerosing pancreatitis (LPSP). It is difficult to obtain sufficient tissue sample to diagnose typical histological features of LPSP, such as striform fibrosis and obliterative phlebitis, thus only a specimen of core biopsy or resection can be used for diagnosis in the international consensus diagnostic criteria (ICDC) [5]. AIP is diagnosed according to ICDC on the basis of the serum IgG4 concentration, histological features of LPSP, pancreatic parenchymal and ductal imaging, other organ involvement, and response to steroid therapy. Although a favorable response of steroid therapy is a characteristic feature of AIP [6,7,8] and a steroid trial is a useful method for the diagnosis of AIP, steroid therapy is potentially associated with adverse effects, especially when a long term use is necessary [9]. The European guideline on IgG4-related digestive disease recommended Endoscopic ultrasonography (EUS)-guided tissue acquisition for the histological diagnosis of AIP and for exclusion of pancreatic carcinoma [10]

Endoscopic ultrasonography-guided fine needle aspiration (EUS-FNA) is an established technique of sample acquisition from the pancreas for cytological evaluation [11,12,13,14]. Novel needles have been developed with side holes or dedicated tip designs to obtain adequate tissue samples for histological evaluation and are called EUS-guided fine needle biopsy (EUS-FNB) needles [15,16,17,18,19,20,21,22,23,24]. Recently, the diagnostic yield of EUS-FNA for AIP has been reported [25,26,27,28,29,30,31,32,33,34,35], and EUS-FNB is expected to provide the better yield [25,36].

However, most of these reports only discussed the sensitivity of EUS-FNA/B for histological diagnosis, and the role of EUS-FNA/B in the diagnostic flow of AIP is not fully elucidated. In the present study, we investigated the contribution of EUS-FNA/B to the diagnosis of AIP.

## 2. Materials and Methods

We retrospectively reviewed medical records, pathological reports, and radiological images of consecutive patients who were clinically suspected with type 1 AIP and underwent EUS-FNA/B at The University of Tokyo hospital between July 2013 and August 2019. Patients were queried from our prospectively maintained database of EUS-FNA. Clinical diagnosis before EUS-FNA was recorded by endosonographers based on the radiological findings with or without serum IgG4 levels. We evaluated the contribution of EUS-FNA/B to the diagnosis of AIP in the diagnostic flow. We used the diagnostic criteria proposed by Japan Pancreas Society in 2018 (JPS 2018) [37], and the final diagnosis was confirmed by follow-up of 6 months or longer.

All EUS-FNA/B were performed under conscious sedation, and the needle size or type and the amount of suction were selected at the discretion of the endoscopist. We classified the needles with side holes or dedicated tip design for tissue acquisition as FNB needles. EUS-FNA was performed using 25-, 22-, and 19-gauge normal FNA needles. EUS-FNB was performed using a 22-gauge Franseen needle, 19-gauge reverse-bevel, and 20-gauge forward-bevel needle.

We reviewed pathological reports of EUS-FNA/B and surveyed 4 typical features of LPSP: prominent lymphoplasmacytic infiltration along with fibrosis, abundant (>10 cells/high-power microscopic field) IgG4-positive plasma cells, storiform fibrosis, and obliterative phlebitis. These histological characteristics of LPSP on biopsy tissues were defined by Notohara et al. [38]. Being found positive for 3 or 4 out of these 4 features was defined as pathologically definite, and positive for 2 was defined as pathologically possible.

We evaluated the contribution of EUS-FNA/B to the diagnosis of AIP prior to the introduction of steroid therapy, because it is noted in the diagnostic criteria [37] that facile diagnostic treatment with steroids should be strictly avoided. Contribution of EUS-FNA/B was assessed in patients with diffuse enlargement of the pancreas and those with focal enlargement, separately. The diagnostic yield of EUS-FNA and that of EUS-FNB was compared using Fisher’s exact test or χ^2^ test, and *p* values <0.05 were considered statistically significant. Statistical analyses were performed using statistical software R 3.3.2 (R Foundation for Statistical Computing, Vienna, Austria). 

The study was conducted in accordance with the Declaration of Helsinki. The local ethical committee approved this study (No. 2058) and waived informed consent due to its retrospective nature.

## 3. Results

### 3.1. Patients Characteristics and Final Diagnosis 

A total of 53 patients with clinical suspicion of type 1 AIP were included in this analysis. Patients’ characteristics are described in Table 1. Twenty-two patients (41.5%) showed diffuse enlargement of the pancreas and 31 (58.5%) showed focal enlargement. Locus of the target of EUS-FNA/B among patients with diffuse enlargement was the pancreatic head in 7 patients (32%), the body in 11 (50%), and the tail in 4 (18%). FNA needles were used in 21 patients (19-gauge standard needle in 12- 22-gauge in 7, and 25-gauge in 2); FNB needles, in 32 (22-gauge Franseen needle in 17, 19-gauge reverse-bevel needle in 3, and 20-gauge forward-bevel needle in 12). The median number of passes was four (1-7). Rapid on-site cytologic evaluation was not performed. The final diagnosis was as follows: 32 patients with definitive AIP, 2 with probable AIP, 1 with possible AIP, and 18 with mass-forming focal pancreatitis. There was no patient with pancreatic cancer in this study cohort. 

### 3.2. Contribution of EUS-FNA/B to Diagnosis of AIP

The diagnostic flow of AIP is shown in Figure 1. Among patients with diffuse enlargement of the pancreas, there was no change in the diagnosis after considering the pathological findings of EUS-FNA/B, compared with the diagnosis based on pancreatic imaging, serum IgG4, and other organ involvement (Figure 1A). Thus, EUS-FNA/B provided no contribution to diagnosis of diffuse type AIP.

On the other hand, the diagnosis changed in 12 patients (38.7%) after considering the pathological findings of EUS-FNA/B among patients with focal enlargement of the pancreas: a probable to definitive diagnosis in 4, unspecified to definitive in 3, and unspecified to probable in 5 (Figure 1B). All of these 12 patients were diagnosed as having definitive or probable AIP before the introduction of steroid treatment. Of the five patients whose diagnosis changed from unspecified to probable AIP after EUS-FNA/B, four received steroid therapy, and all patients responded to steroids and were eventually confirmed as having definite AIP.

Comparison of diagnoses with and without considering the pathological findings of EUS-FNA/B is shown in Figure 2, including response of steroid therapy. Without pathological results by EUS-FNA/B, 26 patients were diagnosed as definitive AIP and 5 were diagnosed as probable. When considering the pathological findings of EUS-FNA/B, we could diagnose 32 patients with definitive AIP and 2 with probable. Three patients might not have had the diagnosis of AIP if EUS-FNA/B had not been performed.

### 3.3. Comparison of the Diagnostic Yield of EUS-FNA/B

Among 34 patients diagnosed with definitive or probable AIP, we compared the diagnostic yield of EUS-FNA (*n* = 14) and EUS-FNB (*n* = 20). Figure 3A shows the comparisons of the sensitivity for each typical feature of LPSP. EUS-FNB provided significantly higher sensitivity for prominent lymphoplasmacytic infiltration along with fibrosis (100% vs. 64.3%, *p* = 0.007), abundant IgG4-positive plasma cells (60.0% vs. 21.4%, *p* = 0.038), and obstructive phlebitis (70.0% vs. 7.1%, *p* <0.001). Although there was no statistically significant difference in storiform fibrosis, only EUS-FNB disclosed storiform fibrosis in threepatients (15%).

Among patients who underwent EUS-FNA, no one was diagnosed as pathologically definitive, and only four were diagnosed as pathologically possible. Meanwhile, among patients who underwent EUS-FNB, 10 were diagnosed as pathologically definitive and 7 were diagnosed as pathologically possible. Thus, EUS-FNB provided a significantly higher diagnostic yield than EUS-FNA (Figure 3B).

Among FNB needles, we further compared the diagnostic yield of 22-gauge Franseen needles (*n* = 10) and that of 20-gauge forward-bevel needles (*n* = 7). A pathologically possible diagnosis was obtained in 5 patients by Franseen needles and in only 1 by forward-bevel needles (50.0% vs. 14.3%, *p* = 0.304), and a pathologically definite diagnosis was obtained in 5 by Franseen needles and in 3 by forward-bevel needles (50.0% vs. 42.9%, *p* = 1).

### 3.4. Adverse Events of EUS-FNA/B

Adverse events of EUS-FNA/B are shown in Table 2. Both patients complicated by pancreatic fistula were treated conservatively. All of the four patients who developed adverse events had focal enlargement of the pancreas and underwent EUS-FNB. Although there was no significant difference in the adverse event rates between EUS-FNA and FNB (0% vs. 12.5%, *p* = 0.143), there were no adverse events in patients who underwent EUS-FNA. Among the four patients with adverse events of EUS-FNB, however, pathological findings were helpful in the diagnosis of AIP in two of them—possible to definitive AIP in one, and from an unspecified diagnosis to probable AIP in another.

## 4. Discussion

In this retrospective analysis on the role of EUS-FNA/B for the diagnosis of AIP, we revealed that EUS-FNA/B effectively contributed to diagnosis prior to the introduction of steroid therapy in 40% of patients with focal enlargement of the pancreas, though in none who had AIP with diffuse enlargement. Finally, 10% of patients with focal enlargement of the pancreas could not be diagnosed as AIP without the pathological findings by EUS-FNA/B, even after a positive response to steroid therapy. EUS-FNB provided a higher diagnostic yield than EUS-FNA, at the cost of potential increase in adverse event rates.

Several studies reported the diagnostic difficulty of AIP by EUS-FNA with the limited sensitivity of 0–19% for pathologically definite diagnosis of LPSP, because EUS-FNA could not provide sufficient tissue samples for histological evaluation [26,27,33,34,35]. Iwashita et al. reported the improved sensitivity of 43% using a 19-gauge FNA needle [29]. Kanno et al. emphasized the importance of quick needle motion for adequate tissue acquisition, and they reported the sensitivity of 41–56% even using 22-gauge FNA needles [30,31]. Recently, Kurita et al. documented for the first time the utility of EUS-FNB using a Franseen needle in the diagnosis of AIP, with a high sensitivity of 56% for pathologically definite diagnosis of LPSP [36].

The primary endpoint of the above studies was sensitivity for pathological diagnosis of AIP by EUS-FNA/B. However, different from pancreatic cancer, the diagnosis of AIP is based on the combination of clinical and pathological features as well as response to steroid therapy. Only a few studies have evaluated the contribution of EUS-FNA/B to the diagnosis of AIP. Additional diagnosis of LPSP by histology ranged from 55.6% to 94.7% in patients without diagnosis of AIP according to the ICDC prior to histological evaluation [26,34,36]. According to the recent multicentric Italian survey on daily practice for AIP, EUS-FNA/B was judged diagnostic for AIP in 75.4% [39]. However, it has not been elucidated who can benefit from EUS-FNA/B in the diagnosis of AIP. Our study revealed that EUS-FNA/B provided useful histological information and changed the diagnosis only in patients with focal enlargement of the pancreas, but not in those with diffuse enlargement. This is consistent with the recent multicentric study by Notohara et al. [40]. They also evaluated the usefulness of histological diagnoses in the context of the diagnostic criteria and illustrated the diagnostic algorithms. They reported that the histological diagnosis contributed to the final diagnosis in only 1 out of 28 cases with diffuse pancreatic swelling, but the histological diagnosis was essential in 25 out of 45 cases with focal swelling. Thus, EUS-FNA/B should be performed in patients with clinical suspicion of AIP with focal enlargement.

In this study, we used the Japanese diagnostic criteria for AIP [37] because it is described in the ICDC that AIP can be diagnosed only on resection or core biopsy specimen. When using the ICDC, we also confirmed that EUS-FNA/B provided significant contribution only among patients with focal disease. Among 31 patients with focal enlargement of the pancreas, 8 were diagnosed with definite type 1 AIP and 23 were diagnosed with unspecified pancreatitis based on pancreatic imaging, serum IgG4 levels, and other organ involvement. After considering the pathological findings by EUS-FNA/B, diagnosis changed in two patients (6.4%), from an unspecified diagnosis to definite AIP. Thus, the contribution of EUS-FNA/B in AIP diagnosis was estimated to be less when using the ICDC as opposed to the JPS2018. This is because histology of the pancreas is originally regarded as less important in the ICDC.

In our analysis, EUS-FNB provided a higher diagnostic yield than EUS-FNA. Sensitivity for obstructive phlebitis was much higher in EUS-FNB compared with EUS-FBA (70.0% vs. 7.0%, *p* <0.001). Fifty percent of patients were diagnosed with pathologically definite LPSP in patients who underwent EUS-FNB, though none were diagnosed of those who underwent EUS-FNA (50.0% vs. 0%, *p* = 0.002). FNB needles can acquire a larger amount of tissue with better preservation of tissue architecture than FNA needles [17,19,41,42]. Thus, special staining, such as Elastica van Gieson staining and IgG4 immunostaining, can be performed and is speculated to improve diagnostic yield of EUS-FNB. Notohara et al. referred to the importance of elastic stain on the evaluation of obliterative phlebitis in the guidance for diagnosing AIP with biopsy tissues [38]. According to two recent randomized trials, end-cutting needles can acquire a larger amount of tissue with better preservation of tissue architecture than side-fenestrated needles in autoimmune pancreatitis and for the sampling of solid pancreatic lesions [36,43]. In our study, the number of cases using each needle was too small to detect the statistical significance, though.

As for procedure-related adverse events, EUS-FNB was associated with a higher adverse event rate than EUS-FNA in our study, though this was not statistically significantly (20.0% vs. 0%, *p* = 0.143). Previous studies comparing EUS-FNA and EUS-FNB for solid pancreatic mass reported no significant difference in adverse event rates [16,19,44]. Since the adverse event rate of EUS-FNB in our cohort appeared too high, further investigation with a larger cohort is warranted.

The limitations of our study are the retrospective nature and the small number of patients enrolled. The selection of the needles was at the discretion of the endoscopist.

## 5. Conclusions

In conclusion, EUS-FNA/B was helpful in the diagnosis of focal type AIP, and steroid therapy could be introduced after the diagnosis was confirmed. EUS-FNB provided a higher diagnostic yield than EUS-FNA in AIP.

## Figures and Tables

**Figure 1 diagnostics-10-00954-f001:**
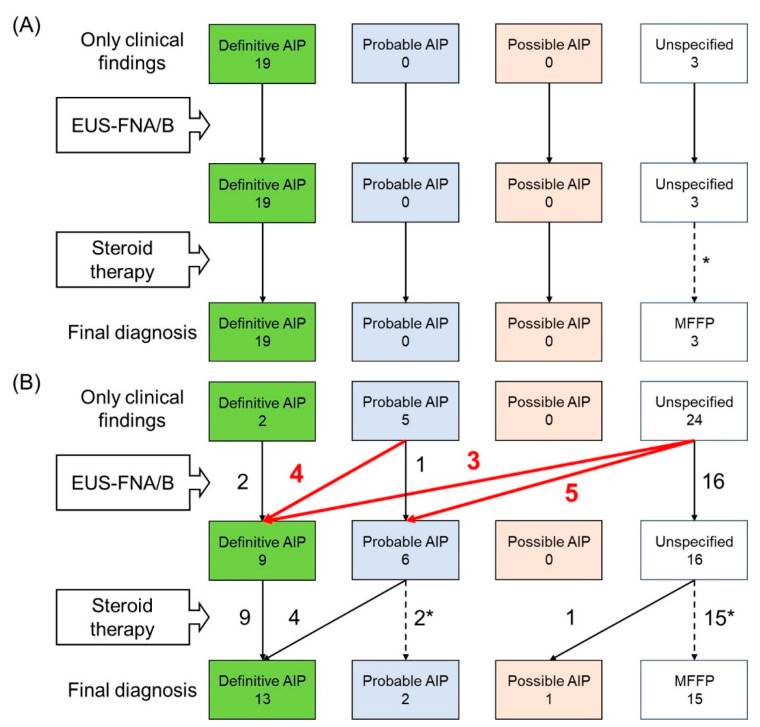
Diagnostic flow of autoimmune pancreatitis according to Japanese Clinical Diagnostic Criteria for Autoimmune Pancreatitis 2018 in patients with diffuse enlargement of the pancreas (**A**) and in those with focal enlargement (**B**). AIP, autoimmune pancreatitis; EUS-FNA/B, endoscopic ultrasonography-guided fine needle aspiration/biopsy; MFFP, mass-forming focal pancreatitis. * They did not receive steroid therapy.

**Figure 2 diagnostics-10-00954-f002:**
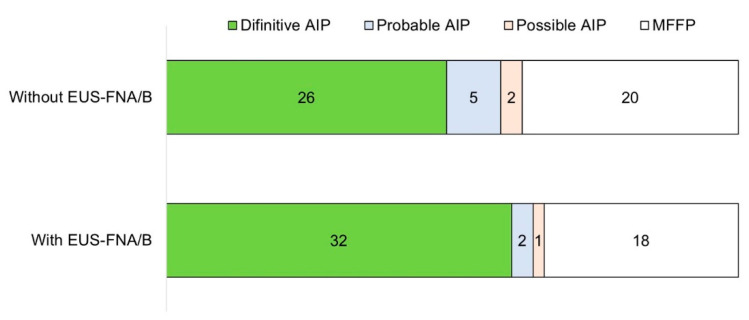
Comparison of diagnosis with and without considering the pathological findings of EUS-FNA/B. EUS-FNA/B, Endoscopic ultrasonography-guided fine needle aspiration/biopsy.

**Figure 3 diagnostics-10-00954-f003:**
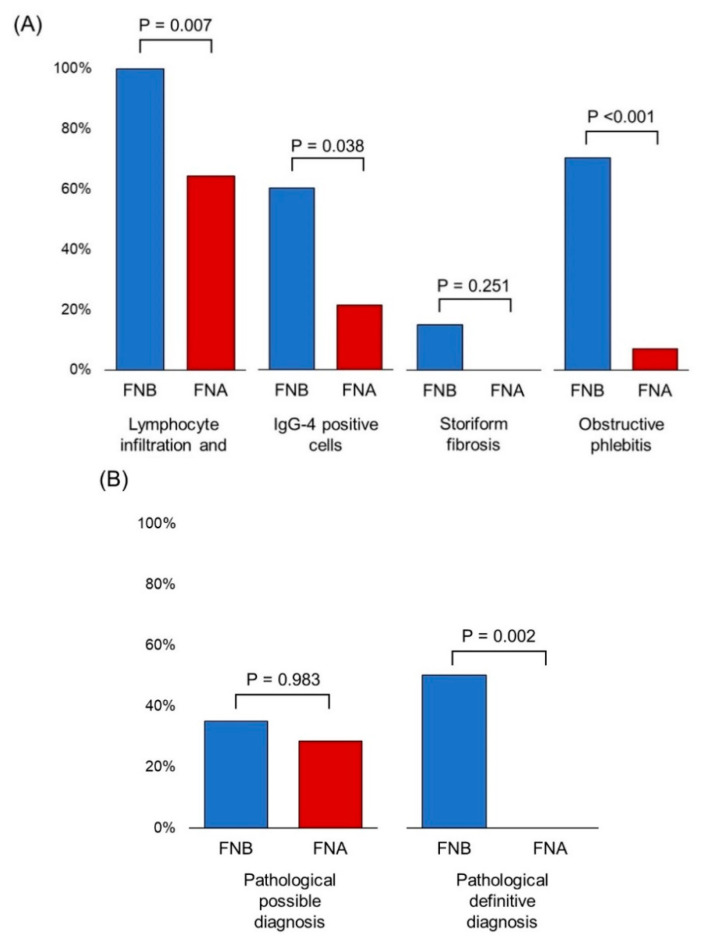
Comparison of the diagnostic yield of EUS-FNA/B among patients with probable and definitive AIP. (**A**) Comparison of the sensitivity for each typical histological finding of lymphoplasmacytic sclerosing pancreatitis. (**B**) Comparison of the sensitivity for pathological diagnosis. AIP, autoimmune pancreatitis; EUS-FNA/B, endoscopic ultrasonography-guided fine needle aspiration/biopsy.

**Table 1 diagnostics-10-00954-t001:** Patients characteristics and final diagnosis.

	*n* = 53
Sex: male	30 (56.6%)
Age * (year)	68 (17–83)
Enlargement of the pancreas: diffuse/focal	22/31 (41.5%/58.5%)
Elevated levels of serum IgG4 (≥ 135 mg/dl)	33 (62.3%)
Other organ involvement	21 (40.0%)
Locus of the target of EUS-FNA/B	
Head	14 (26.4%)
Body	25 (47.2%)
Tail	14 (26.4%)
Diameter of the lesion * (mm)	25 (20–36)
Needles: FNA/FNB	21/32 (39.6%/60.4%)
Final diagnosis	
Definitive AIP	32 (60.4%)
Probable AIP	2 (3.8%)
Possible AIP	1 (1.9%)
Mass-forming focal pancreatitis	18 (34.0%)

Values are expressed as numbers (%) otherwise indicated. * Values are expressed as medians (ranges). EUS-FNA/B, endoscopic ultrasonography-guided fine needle aspiration/biopsy; AIP, autoimmune pancreatitis.

**Table 2 diagnostics-10-00954-t002:** Adverse events of endoscopic ultrasonography-guided fine needle aspiration/biopsy.

	Enlargement of the Pancreas	Needle	Adverse Event	Diagnosis without EUS-FNA/B	Diagnosis with EUS-FNA/B
1	Focal	FNB 22-gauge Franseen	Pancreatic fistula	Unspecified diagnosis	MFFP
2	Focal	FNB 22-gauge Franseen	Pancreatic fistula	Unspecified diagnosis	Probable AIP
3	Focal	FNB 20-gauge forward-bevel	Mild pancreatitis	Unspecified diagnosis	MFFP
4	Focal	FNB 20-gauge forward-bevel	Mild pancreatitis	Possible AIP	Definitive AIP

EUS-FNA/B, endoscopic ultrasonography-guided fine needle aspiration/biopsy; MFFP, mass-forming focal pancreatitis; AIP, autoimmune pancreatitis.

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
