# Peer review of "Role of Endoscopic Ultrasonography-Guided Fine Needle Aspiration/Biopsy in the Diagnosis of Autoimmune Pancreatitis"

_diagnostics, 2020, doi:10.3390/diagnostics10110954_

Round 1

Reviewer 1 Report

This study demonstrated that EUS sampling, particularly EUS-FNB is useful for the diagnosis of focal AIP. Differently, as expected, EUS-sampling should be avoided in diffuse form because does not implement the diagnosis.

Despite I agree with the key points of this study, I have some major methodological concerns:

Technical details of FNA and FNB should be added (e.g., needle type used, number of passes, use of ROSE, site of biopsy in cases of diffuse pancreatic enlargement)

Patients’ selection is not clear. What exactly mean “clinically suspected for AIP”? Which database did you query? Is there an AIP registry available at your site? In a retrospective study, it is of paramount importance to clearly state how the analyzed population was selected.

Pathological findings of AIP on biopsy specimens are not defined yet. You should state the definitions of pathological criteria used to define the features of LPSP. From this point of view, the pathological guide recently published (Notohara K, et al. Guidance for diagnosing autoimmune pancreatitis with biopsy tissues. Pathol Int. 2020 Aug 6. doi: 10.1111/pin.12994) could be of help.

Please clarify how pancreatic fistulas were managed.

Author Response

Point by Point Response to the Reviewers’ Comments

[In our manuscript, changed or added sentences are colored in red]

Reviewer #1: <Comments to Author>

  1. Technical details of FNA and FNB should be added (e.g., needle type used, number of passes, use of ROSE, site of biopsy in cases of diffuse pancreatic enlargement)

[Answer]

We appreciate your comments. We added the following sentences to Results.

Page 3, line 96:

Twenty-two patients (41.5%) showed diffuse enlargement of the pancreas, and 31 (58.5%) showed focal enlargement. Locus of the target of EUS-FNA/B among patients with diffuse enlargement was the pancreatic head in 7 patients (32%), the body in 11 (50%), and the tail in 4 (18%). FNA needles were used in 21 patients (19-gauge standard needle in 12, 22-gauge in 7, and 25-gauge in 2); and FNB needles, in 32 (22-gauge Franseen needle in 17, 19-gauge reverse-bevel needle in 3, and 20-gauge forward-bevel needle in 12). Median number of passes was 4 (1-7). Rapid on-site cytologic evaluation was not performed.

  1. Patients’ selection is not clear. What exactly mean “clinically suspected for AIP”? Which database did you query? Is there an AIP registry available at your site? In a retrospective study, it is of paramount importance to clearly state how the analyzed population was selected.

[Answer]

We appreciate your comments. We used the EUS-FNA database at the University of Tokyo. We added the following sentences to Methods.

Page 2, line 69:

Patients were queried from our prospectively maintained database of EUS-FNA. Clinical diagnosis before EUS-FNA was recorded by endosonographers based on the radiological findings with or without serum IgG4 levels.

  1. Pathological findings of AIP on biopsy specimens are not defined yet. You should state the definitions of pathological criteria used to define the features of LPSP.

[Answer]

We appreciate your comments. We revised our paper according to your recommendations.

Page 2, line 80: 

We reviewed pathological reports of EUS-FNA/B, and surveyed 4 typical features of LPSP: prominent lymphoplasmacytic infiltration along with fibrosis, abundant (>10 cells/high-power microscopic field) IgG4-positive plasma cells, storiform fibrosis, and obliterative phlebitis. These histological characteristics of LPSP on biopsy tissues were defined by Notohara et al (ref).

  1. Please clarify how pancreatic fistulas were managed.

[Answer]

We appreciate your comments. We revised our paper according to your recommendations.

Page 6, line 160: 

Adverse events of EUS-FNA/B are shown in Table 2. Both two patients complicated by pancreatic fistula were treated conservatively.

Reviewer #2: <Comments to Author>

Minor point

  1. The results are not entirely novel, as the usefulness of FNB in this context has been previously reported. While many of such studies are cited by the authors, I think they should discuss also:

the usefulness of EUS-FNB in the context of daily clinical practice where ICDC criteria are seldom eployed (Barresi et al. UEGJ 2020)

comment on the recent European guidelines on IgG4RD (Lohr et al. UEGJ 2020) and on the "Guidance for diagnosing autoimmune pancreatitis with biopsy tissues" (Notohara  Pathol Int 2020)

cite the study by Notohara on " Efficacy and limitations of the histological diagnosis of type 1 autoimmune pancreatitis with endoscopic ultrasound-guided fine needle biopsy (Pancreatology 2020)

[Answer]

We appreciate your comments. We added the following sentences.

Page 2, line 54:

The European guideline on IgG4-related digestive disease recommended EUS-guided tissue acquisition for the histological diagnosis of AIP and for exclusion of pancreatic carcinoma (ref: Lohr UEGJ 2020).

Page 7, line 194:

Additional diagnosis of LPSP by histology ranged from 55.6 to 94.7% in patients without diagnosis of AIP according to the ICDC prior to histological evaluation [26,34,36]. According to the recent multicentric Italian survey on daily practice for AIP, EUS-FNA/B was judged diagnostic for AIP in 75.4%. (ref: Barresi UEGJ 2020)

Page 7, line 195:

However, it has not been elucidated who can benefit from EUS-FNA/B in the diagnosis of AIP. Our study revealed that EUS-FNA/B provided useful histological information and changed the diagnosis only in patients with focal enlargement of the pancreas, but not in those with diffuse enlargement. This is consistent with the recent multicentric study by Notohara et al. (ref: Notohara Pancreatology 2020) They also evaluated the usefulness of histological diagnoses in the context of the diagnostic criteria and illustrated the diagnostic algorithms. They reported that the histological diagnosis contributed to the final diagnosis only in 1 out of 28 cases with diffuse pancreatic swelling, but the histological diagnosis was essential in 25 out of 45 cases with focal swelling.

Page 8, line 220:

Thus, special staining, such as EVG staining and IgG4 immunostaining, can be performed and is speculated to improve diagnostic yield of EUS-FNB. Notohara et al referred to the importance of elastic stain on the evaluation of obliterative phlebitis in the guidance for diagnosing AIP with biopsy tissues (ref: Notohara Patho Int 2020).

  1. The authors should defintely give more details about the exact needles used here (22-gauge Franseen needle or a 20-gauge forward-bevel needle ??).

[Answer]

We added the following sentences to Methods and Results.

Page 2, line 77:

EUS-FNA was performed using 25, 22, and 19-gauge normal FNA needles. EUS-FNB was performed using 22-gauge Franseen needle, 19-gauge reverse-bevel, and 20-gauge forward-bevel needle.

Page 3, line 97:

Locus of the target of EUS-FNA/B among patients with diffuse enlargement was the pancreatic head in 7 patients (32%), the body in 11 (50%), and the tail in 4 (18%). FNA needles were used in 21 patients (19-gauge standard needle in 12, 22-gauge in 7, and 25-gauge in 2); and FNB needles, in 32 (22-gauge Franseen needle in 17, 19-gauge reverse-bevel needle in 3, and 20-gauge forward-bevel needle in 12). Median number of passes was 4 (1-7). Rapid on-site cytologic evaluation was not performed.

  1. The rate of adverse events is not that low. Which needle was used in those cases?

[Answer]

20-gauge forward bevel needles were used in 2 patients with mild pancreatitis, and 22-gauge Franseen needles were used in 2 with pancreatic fistula. We added the details about the needles to Table 2.

  1. I am surprised there are NO CASES of CANCER among the whole study population with suspect of AIP. Comment please.

[Answer]

In most of the cases, the serum levels of IgG4 were already known at the time of EUS-FNA/B. Thus, there were no cases of cancer among this small study cohort.

Reviewer 2 Report

This is an interesting study evaluating the diagnostic relevance of FNB for the diagnosis of type I AIP in patients presenting with a clinical/radiologial suspect either because of "mass forming" disease or for a "diffuse form".

I do not have major comments as the results are clearly presented.

The results are not entirely novel, as the usefulness of FNB in this context has been previously reported. While many of such studies are cited by the authors, I think they should discuss also:

  • the usefulness of EUS-FNB in the context of daily clinical practice where ICDC criteria are seldom eployed (Barresi et al. UEGJ 2020)
  • comment on the recent European guidelines on IgG4RD (Lohr et al. UEGJ 2020) and on the "Guidance for diagnosing autoimmune pancreatitis with biopsy tissues" (Notohara  Pathol Int 2020)
  • cite the study by Notohara on " Efficacy and limitations of the histological diagnosis of type 1 autoimmune pancreatitis with endoscopic ultrasound-guided fine needle biopsy (Pancreatology 2020)

The authors should defintely give more details about the exact needles used here (22-gauge Franseen needle or a 20-gauge forward-bevel needle ??).

The rate of adverse events is not that low. Which needle was used in those cases?

I am surprised there are NO CASES of CANCER among the whole study population with suspect of AIP. Comment please.

Author Response

Point by Point Response to the Reviewers’ Comments

[In our manuscript, changed or added sentences are colored in red]

Reviewer #2: <Comments to Author>

  1. The results are not entirely novel, as the usefulness of FNB in this context has been previously reported. While many of such studies are cited by the authors, I think they should discuss also:

the usefulness of EUS-FNB in the context of daily clinical practice where ICDC criteria are seldom eployed (Barresi et al. UEGJ 2020)

comment on the recent European guidelines on IgG4RD (Lohr et al. UEGJ 2020) and on the "Guidance for diagnosing autoimmune pancreatitis with biopsy tissues" (Notohara  Pathol Int 2020)

cite the study by Notohara on " Efficacy and limitations of the histological diagnosis of type 1 autoimmune pancreatitis with endoscopic ultrasound-guided fine needle biopsy (Pancreatology 2020)

[Answer]

We appreciate your comments. We added the following sentences.

Page 2, line 54:

The European guideline on IgG4-related digestive disease recommended EUS-guided tissue acquisition for the histological diagnosis of AIP and for exclusion of pancreatic carcinoma (ref: Lohr UEGJ 2020).

Page 7, line 194:

Additional diagnosis of LPSP by histology ranged from 55.6 to 94.7% in patients without diagnosis of AIP according to the ICDC prior to histological evaluation [26,34,36]. According to the recent multicentric Italian survey on daily practice for AIP, EUS-FNA/B was judged diagnostic for AIP in 75.4%. (ref: Barresi UEGJ 2020)

Page 7, line 195:

However, it has not been elucidated who can benefit from EUS-FNA/B in the diagnosis of AIP. Our study revealed that EUS-FNA/B provided useful histological information and changed the diagnosis only in patients with focal enlargement of the pancreas, but not in those with diffuse enlargement. This is consistent with the recent multicentric study by Notohara et al. (ref: Notohara Pancreatology 2020) They also evaluated the usefulness of histological diagnoses in the context of the diagnostic criteria and illustrated the diagnostic algorithms. They reported that the histological diagnosis contributed to the final diagnosis only in 1 out of 28 cases with diffuse pancreatic swelling, but the histological diagnosis was essential in 25 out of 45 cases with focal swelling.

Page 8, line 220:

Thus, special staining, such as EVG staining and IgG4 immunostaining, can be performed and is speculated to improve diagnostic yield of EUS-FNB. Notohara et al referred to the importance of elastic stain on the evaluation of obliterative phlebitis in the guidance for diagnosing AIP with biopsy tissues (ref: Notohara Patho Int 2020).

  1. The authors should defintely give more details about the exact needles used here (22-gauge Franseen needle or a 20-gauge forward-bevel needle ??).

[Answer]

We added the following sentences to Methods and Results.

Page 2, line 77:

EUS-FNA was performed using 25, 22, and 19-gauge normal FNA needles. EUS-FNB was performed using 22-gauge Franseen needle, 19-gauge reverse-bevel, and 20-gauge forward-bevel needle.

Page 3, line 97:

Locus of the target of EUS-FNA/B among patients with diffuse enlargement was the pancreatic head in 7 patients (32%), the body in 11 (50%), and the tail in 4 (18%). FNA needles were used in 21 patients (19-gauge standard needle in 12, 22-gauge in 7, and 25-gauge in 2); and FNB needles, in 32 (22-gauge Franseen needle in 17, 19-gauge reverse-bevel needle in 3, and 20-gauge forward-bevel needle in 12). Median number of passes was 4 (1-7). Rapid on-site cytologic evaluation was not performed.

  1. The rate of adverse events is not that low. Which needle was used in those cases?

[Answer]

20-gauge forward bevel needles were used in 2 patients with mild pancreatitis, and 22-gauge Franseen needles were used in 2 with pancreatic fistula. We added the details about the needles to Table 2.

  1. I am surprised there are NO CASES of CANCER among the whole study population with suspect of AIP. Comment please.

[Answer]

In most of the cases, the serum levels of IgG4 were already known at the time of EUS-FNA/B. Thus, there were no cases of cancer among this small study cohort.

Round 2

Reviewer 1 Report

Thank you for amending the manuscript.

I think the paper is now suitable for publication.

I have only another minor request for the Authors. In your study, you used the 22-gauge Franseen needle in 17 patients and the 20-gauge forward-bevel needle in 12. Two recent randomized trials demonstrated end-cutting needles to perform better than side fenestrated one in autoimmune pancreatitis and for the sampling of solid pancreatic lesions (Kurita et al, already cited in your study; and Crinò et al. Gastrointest Endosc. 2020 Sep;92(3):648-658, please add this reference). It would be interesting for the readers to know if you found any difference between these two needles. Please, add this result and discuss it.

Good luck with the publication.

Author Response

Point by Point Response to the Reviewers’ Comments

[In our manuscript, changed or added sentences are colored in blue]

Reviewer #1: <Comments to Author>

Minor

  1. I have only another minor request for the Authors. In your study, you used the 22-gauge Franseen needle in 17 patients and the 20-gauge forward-bevel needle in 12. Two recent randomized trials demonstrated end-cutting needles to perform better than side fenestrated one in autoimmune pancreatitis and for the sampling of solid pancreatic lesions (Kurita et al, already cited in your study; and Crinò et al. Gastrointest Endosc. 2020 Sep;92(3):648-658, please add this reference). It would be interesting for the readers to know if you found any difference between these two needles. Please, add this result and discuss it.

[Answer]

We appreciate your comments. We added the following sentences to Results and discussion.

Page 5, line 152:

Among FNB needles, we further compared the diagnostic yield of 22-gauge Franseen needles (N=10) and that of 20-gauge forward-bevel needles (N=7). Pathologically possible diagnosis was obtained in 5 patients by Franseen needles and in only 1 by forward-bevel needles (50.0% vs. 14.3%, P = 0.304), and pathologically definite diagnosis was obtained in 5 by Franseen needles and in 3 by forward-bevel needles (50.0% vs. 42.9%, P = 1).

Page 8, line 228:

According to two recent randomized trials, end-cutting needles can acquire a larger amount of tissue with better preservation of tissue architecture than side-fenestrated needles in autoimmune pancreatitis and for the sampling of solid pancreatic lesions [36,43]. In our study, the number of cases using each needle was too small to detect the statistical significance, though.
